# Antioxidative and Energy Metabolism-Improving Effects of Maca Polysaccharide on Cyclophosphamide-Induced Hepatotoxicity Mice via Metabolomic Analysis and Keap1-Nrf2 Pathway

**DOI:** 10.3390/nu14204264

**Published:** 2022-10-12

**Authors:** Wenting Fei, Jianjun Zhang, Shuhui Yu, Na Yue, Danni Ye, Yingli Zhu, Ran Tao, Yan Chen, Yawen Chen, Aimin Li, Linyuan Wang

**Affiliations:** 1School of Chinese Materia Medica, Beijing University of Chinese Medicine, Beijing 102401, China; 2School of Traditional Chinese Medicine, Beijing University of Chinese Medicine, Beijing 102401, China; 3Yantai Xinshidai Health Industry Co., Ltd., Yantai 264006, China

**Keywords:** maca polysaccharide, hepatotoxicity, cyclophosphamide, metabolomics, antioxidative, energy metabolism, Keap1-Nrf2

## Abstract

*Lepidium meyenii* Walp. (Maca), as a natural food supplement, has strong antioxidant and energy metabolism-improving characteristics, and Maca polysaccharide (MP) is its effective component. MP has been shown to mitigate liver damage in previous research, and Cyclophosphamide (CYP)-induced hepatotoxicity is also a major concern in clinical practice. We investigated the possible cytoprotective effect of MP on CYP-induced liver injury, and explored its underlying mechanism by analyzing the resulting liver metabolic profiles. MP significantly inhibited increases in serum transaminase, improved pathological changes, reduced oxidative stress, and increased the levels of energy metabolism-related enzymes. Metabolomic analysis showed that MP corrected lipid metabolic problems and regulated the pentose phosphate pathway and acid metabolism, thereby protecting against apoptosis of hepatocytes. The Pearson correlation analysis indicated that antioxidant enzymes and energy metabolism-related enzymes are closely correlated with these differential metabolites. In addition, the upstream Keap1-Nrf2 antioxidant signal transduction pathway was explored to validate the possible mechanism of the cytoprotective effect of MP. In conclusion, MP plays a protective role in CYP-induced hepatotoxicity through these potential metabolic means, where it ameliorates oxidative stress, improves energy metabolism, and restores mitochondrial respiration by regulating the Keap1-Nrf2 signaling pathway, thereby preventing liver damage.

## 1. Introduction

*Lepidium meyenii* Walp. (Maca) is native to the Andes of Peru in South America and grows exclusively at altitudes over 3500 m [1]. In traditional applications, it can be used to make soup or boil milk, and it can be eaten fresh or dried. At the same time, it can also be used to make drinks, cocktails, or coffee [2]. Because of its unique biological activities, Maca has become a functional plant food and traditional herbal medicine, and a large area of Maca has been planted in Yunnan and Tibet of China. The activities mainly come from its active ingredients, such as polysaccharides, alkaloids, or flavonoids [3]. Researches have shown that the rats subjected to the forced swim test that consumed Maca showed higher levels of superoxide dismutase (SOD), and lower levels of catalase (CAT), lactate dehydrogenase (LDH), and lipid peroxidation (LPO), demonstrating its benefits to the body and its role as an adaptive agent [4].

Maca polysaccharide (MP), as the main active component, has been reported to have many medicinal effects, such as sexual dysfunction regulation; antidepressant, antioxidant, and neuroprotective effects; memory enhancement; anti-cancer and anti-inflammatory activities; and so on [5,6]. Our previous research showed that MP has the effects of preventing weight loss, enhancing mitochondrial energy metabolism enzyme activity when taking cyclophosphamide (CYP), and improving spleen deficiency syndrome of traditional Chinese medicine [7]. MP has good immunomodulatory effects on an immunosuppression model [8].

CYP is widely used in clinics. It is a chemotherapeutic drug that can be used to treat different types of tumors [9]. However, along with the curative effects are the fatal side effects of CYP, including hepatotoxicity, nephrotoxicity, and so on [10]. CYP is a cytotoxic oxazophos alkylating agent and a cell cycle-nonspecific drug. CYP has no activity in vitro and is metabolized into phosphoramide nitrogen mustard and acrolein by cytochrome P450 enzyme system in vivo [11]. Phosphamide nitrogen mustard adds alkyl into the guanine base of DNA and inhibits the synthesis of DNA and RNA, thus inhibiting the synthesis of nucleic acids and playing an anti-tumor and immunosuppressive role. Acrolein can increase reactive oxygen species (ROS) and destroy mitochondrial function, subsequently inducing an antioxidant barrier, intensifying LPO and cell damage, and finally, increasing oxidative stress and hepatocyte apoptosis [12]. Acrolein also causes increases in alanine aminotransferase (ALT), aspartate transaminase (AST), LDH, and other enzyme activities in serum.

The activity of antioxidant enzymes and energy metabolism in the liver is related to the process of liver injury. Oxidative damage of lipids can lead to structural damage and dysfunction of cells or organelles because lipids are a component of biofilm [13]. The level of MDA (malondialdehyde) is a sign of lipid peroxidation, indicating the severity of oxidative stress and the attack of free radicals on cells [14,15]. SOD, GSH-Px (glutathione peroxidase), and CAT as antioxidant enzymes can prevent the harmful effects of oxidative stress, and the levels are a sign of oxidative activities in the body [16,17]. A decrease or deficiency in the activity of mitochondrial respiratory chain complexes in liver cells leads to the dysfunction of ATP synthesis and the disorder of energy metabolism in liver cells, in turn leading to a decrease in energy metabolism in the body. Na^+^-K^+^-ATPase and Ca^2+^-Mg^2+^-ATPase are the two most important ion pump enzymes in ATPase.

Substances with antioxidant activity may protect against tissue damage caused by cyclophosphamide. Researches have shown that Maca polysaccharide possesses hepatoprotective activity against hepatic injury induced by alcohol [18]. Thus, we aimed to investigate the possible cytoprotective effect of MP on CYP-induced liver injury, and to observe its capacity for antioxidant and energy metabolism enhancement.

In this study, by analyzing the resulting liver metabolic profiles, we found that MP regulated glycerophospholipid metabolism, arachidonic acid metabolism, the pyrimidine pathway, the pentose phosphate pathway, and amino acid metabolism disorders. The mechanism is related to the signal pathway of Keap1-Nrf2.

## 2. Materials and Methods

### 2.1. Materials

*Lepidium meyenii* (Maca) powder was supplied by Yantai Xinshidai Health Industry Co., Ltd. (Yantai, Shandong, China), lot number: 20160316. The dried Maca powder was extracted twice by water at 20:1 mL/mg liquid-to-solid ratio. The extraction conditions used were a temperature a 70 °C for 1.5 h, and the columns used in the purification process were a DEAE cellulose chromatography column and Sephadex G-100 gel column. The extraction of MP could reach 9.88 mg/g for further separation, and the purity of total sugar was 75.42% and that of protein was 7.73%. A polysaccharide MP with a molecular weight of 10.6 kDa was isolated from Maca, and MP was isolated by the method of Zha [14], which consisted of rhamnose (Rha), glucose (Glc), galactose (Gal), and arabinose (Ara) at a molar ratio of 6.85:1.81:3.21:1. Arabinose was the main ingredient in MP. MP is mainly composed of →4)-α-D-Glcp-(1→, →6)-α-D-Glcp-(1→, →3)-α-D-Glcp-(1→, and β-D-Araf-(1→, with branching at O-6 of →4,6)-α-D-Glcp-(1 → [19].

### 2.2. Reagents

Injectable CYP was supplied by Bioway Co., Ltd. (Shanghai, China); the lot number was 0J418A, Cas: 6055-19-2.

### 2.3. Animals

A total of 48 Kunming male mice (8 weeks old) were obtained from SPF (Biotechnology Co., Ltd., Beijing, China; Permit number: SCXK (Beijing) 2021-0033). The feeding environment of mice was as follows: light and dark cycles of 12 h, an indoor temperature of 23 ± 2 °C, and humidity of 60 ± 5%. The mice were fed with a standard diet and were allowed to drink freely. The animals were adaptively fed for one week before the experiment. The animals used and experimental scheme were approved by the Ethics Committee in Beijing University of Chinese Medicine (BUCM-4-2021082004-3031).

### 2.4. Experimental Design

The mice were divided into 4 groups (Figure 1), with 12 in each group, randomly. Group I (Normal Control): mice received saline for 14 days. Group II (MP Control): mice received MP 750 mg/kg per day for 14 days. Group III (CYP Model): mice received saline for 14 days and intraperitoneal (i.p.) injection with a dose of cyclophosphamide (CYP) at 60 mg/kg per day for a consecutive 3 days from day 12 to day 14. Group IV (MP+CYP): mice were treated with MP at 750 mg/kg per day for 14 days and i.p. CYP 60 mg/kg per day for a consecutive 3 days from day 12 to day 14. All the animals received saline and MP by oral gavage.

The dose of MP used in this study was based on previous experimental data [8]. The modeling dose and period of CYP was also reported to induce hepatotoxicity in previous experimental reports [20].

### 2.5. Bodyweight and Daily Observation

The bodyweight, mental state, food intake, hair gloss, and so on of mice in each group were observed and recorded daily.

### 2.6. Sample Collection

On the 15th day of the study (the end), blood was collected after all the animals were sacrificed. The blood samples were placed in a homogenizer (Saiweier Biotechnology Co., Ltd.) and centrifuged at 1200× *g* at 4 °C for 5 min. They were then stored at −20 °C in a low-temperature refrigerator for further detection and analysis of biochemical indicators, such as liver function transferases. The left lobule of each liver was cut and immersed in 4% formalin. Other parts of the livers were collected immediately and prepared as homogenates for subsequent biochemical analysis, storing the remaining samples at −80 °C in a refrigerator.

### 2.7. Histopathological Analysis of Livers

The left lobule of liver samples was embedded in paraffin according to routine histological methods. We cut the samples to a size of 5.0 μM. Liver slices were observed by using microscope after staining with hematoxylin and eosin (H&E), after which they were photographed for analysis.

### 2.8. Assessment of Liver Function Transferases

The activities of liver enzymes such as ALT and AST were estimated in serum [21]. ALT (lot number: 20210901-20684A) and AST (lot number: 20210901-20732A) kits were from NJBI (Nanjing Jiancheng Bioengineering Institute, Nanjing, China).

### 2.9. Determination of Hepatic Oxidative Stress Markers

We took 200 mg of hepatic tissue homogenate. The changes in concentration of SOD, CAT, GSH-Px, and MDA in each group were measured by the colorimetric method using an automatic biochemical instrument [22,23,24], in accordance with the kit instructions. The SOD (lot number: 20210612), CAT (lot number: 20210615), GSH-Px (lot number: 20210720), and MDA (lot number: 20210610) kits were purchased from NJBI.

### 2.10. Determination of ATPases in Liver Tissue

Briefly, 0.1 g mL^−1^ homogenates of the liver were prepared for biochemical assays by isotonic physiological saline. The homogenates were centrifuged at 1000 rpm with a homogenizer (Saiweier Biotechnology Co., Ltd.) for 5 min. The activities of Na^+^-K^+^-ATPase and Ca^2+^-Mg^2+^-ATPase were detected from the supernatant [25,26].

F_0_F_1_-ATPase, also known as ATP synthase, is sensitive to oligomycin. In the continuous cycle reaction system of pyruvate kinase (PK) and LDH with oligomycin participation, changes in nicotinamide adenine dinucleotide (NADH) absorption peaks during ATP hydrolysis corresponding to ATP synthesis were measured. The amount of enzyme required to oxidize 1 mol NADH per minute was an active unit, and the specific detection was carried out according to the kit instructions [27]. The F_0_F_1_-ATPase activity spectrometric quantitative detection kit was provided by GENMED Mitochondrial Respiratory Chain Complex V (Shanghai Gemei Gene Pharmaceutical Technology Co., Ltd., lot number: GMS50083).

### 2.11. Hepatic Metabolomic Analysis

#### 2.11.1. Liver Sample Metabolite Extraction and QC Sample Preparation

We took 25 mg of liver sample from mice and added it to the mixed extract containing methanol, acetonitrile, and water. Then, we homogenized the sample and conducted three ultrasounds. The supernatant was transferred to a glass bottle for detection after centrifugation. The samples used for quality control (QC) came from the mixing of all samples.

#### 2.11.2. LC-MS/MS Analysis

The UHPLC system (Vanquish) from Thermo Fisher Scientific was used for LC-MS/MS analyses. Ammonium acetate and ammonia hydroxide together comprised the mobile phase, and the automatic sampler was operated at 4 °C with 2 μL injections. The ESI source conditions and information collection system mode were selected from previous relevant studies [28].

#### 2.11.3. Data Preprocessing and Multivariate Data Analysis

We used R language to preprocess the original data and annotate the metabolites based on the BiotreeDB database. Principal component analysis (PCA) and orthogonal projection to latent structure discriminate analysis (OPLS-DA) were performed with SIMCA software (version 16.0.2, Sartorius Stedim Data Analytics AB, Sweden).

#### 2.11.4. Metabolite Identification

The different metabolites between the Normal Control group (Group I), CYP Model group (Group III), and MP+CYP group (Group IV) were filtered out with variable importance in projection (VIP) > 1 and *p* < 0.05. The key metabolic pathways were identified with MetaboAnalyst 5.0 (https://www.metaboanalyst.ca/, accessed on 8 July 2022) based on the database of the Kyoto Encyclopedia of Genes and Genomes (KEGG) [29].

### 2.12. Quantitative Real-Time PCR Analysis

The gene expression levels of nuclear factor (erythroid-derived 2)-like 2 (Nrf2), Kelch-like ECH-associated protein 1 (Keap1), heme oxygenase 1 (HO-1), glutamate cysteine ligase catalytic subunit (Gclc), and NAD (P) H dehydrogenase quinone 1 (Nqo1) were detected by real-time fluorescence quantitative polymerase chain reaction (RT-qPCR).

Total RNA was extracted from liver tissues of 8 mice in each group according to the instructions for the Trizol Reagent reagent, and then the content and purity of total RNA were detected. Then, 3 µg of total RNA was retrieved and amplified by PCR in a 50 UL reaction system according to the kit instructions. Meanwhile, a dissolution curve was generated to confirm whether there was a single peak. PCR reaction and data acquisition were performed with the Light Cycler 2.0 system, and the cyclic threshold (CT) was recorded. The gene expression was calculated by the relative quantitative formula 2^−ΔΔCT^. The value of Δ(Ct) was equal to the value of the target gene (Ct) minus GAPDH (Ct). The expressions of Nrf2, Keap1, HO-1, Gclc, and Nqo1 in the liver were measured. The primer sequence was designed with primer and probe design software Primer 5.0. The annealing temperature was 60 °C. The primers used and the internal reference GAPDH primers are shown in Table 1.

### 2.13. Determination of Protein Expression Level Using Western Blotting

The total proteins of the liver were extracted and then the protein concentration was determined with a BCA protein assay kit (cat: G2026). The membranes were blocked for 1 h and incubated with anti-Nrf2 rabbit pAb (cat: 16396-1-AP), anti-Keap1 rabbit pAb (cat: 10503-1-AP), anti-HO-1 rabbit pAb (cat: 10701-1-AP), anti-Gclc rabbit pAb (cat: 12601-1-AP), and anti-Nqo1 rabbit pAb (cat: AB80588) at 4 °C overnight [30]. The blots were developed with an enhanced chemiluminescence (ECL) kit (cat: G2014). The BCA and ECL kits and all of the antibodies were purchased from Servicebio (Wuhan, China).

### 2.14. Statistical Analysis

The data of each group were analyzed by GraphPad Prism 9 (GraphPad Software, Inc., San Diego, USA) and SPSS 20.0 (SPSS Inc., IBM Corp., NY, USA). All data are expressed as means ± SEM. *p* < 0.05 was considered as statistical significance. The correlation between the biochemical indices and the potential biomarkers was carried out by the Person correlation analysis and R Programming Language (4.1.0).

## 3. Results

### 3.1. Effects of Maca Polysaccharide on General Conditions and Body Weights of CYP-Treated Mice

Maca polysaccharide can modify the poor conditions and delay the body weight loss induced by CYP in mice (Figure 2). The initial body masses of all groups were the same and showed no statistical difference. The body weight was increased by MP supplement in normal mice, but the mice in the CYP Model group showed obvious weight loss and symptoms of fatigue, laziness, and crouching. These same symptoms were alleviated in the MP+CYP administration group. CYP was injected intraperitoneally from the 12th day for modeling. The weight of the CYP Model mice was significantly lower than that of normal control mice from day 12 to day 14 after injection (*p* < 0.01), while the weight of MP+CYP mice was lower than that of normal control mice due to CYP treatment but significantly higher than that of CYP Model mice (*p* < 0.01).

### 3.2. Maca Polysaccharide Alleviates Liver Histopathological Changes Induced by CYP

The structure of hepatic lobules was normal and the outline was clear in normal control mice and MP control mice (Figure 3A1,A2,B1,B2). The hepatocytes were arranged around the middle vein radially, with clear nucleolus, normal size and shape, no inflammatory cell infiltration, and no edema or expansion of the hepatic sinuses. In contrast, the outline of hepatic lobules was blurred; the structure was disordered; the structure of hepatocytes was abnormal; vacuolar degeneration, nuclear pyknosis, and central vein congestion were obvious; and there was significant inflammatory cell infiltration in the model group (Figure 3C1,C2). However, MP supplementation significantly alleviated the liver histopathological changes induced by CYP (Figure 3D1,D2). Compared with the model group, there was only a small amount of inflammatory cell infiltration and central venous congestion in the liver tissue of the MP group, indicating that MP has a good protective effect against CYP-induced hepatotoxicity.

### 3.3. Maca Polysaccharide Reduces the Levels of Serum Transaminase in CYP-Induced Hepatotoxicity Mice

Serum transaminase (ALT, AST) is an important index reflecting the extent of hepatocytes injury and necrosis (Figure 4). The levels of ALT and AST in serum of the CYP Model mice increased significantly when compared with normal mice (*p* < 0.001, *p* < 0.0001). Supplementation with MP had no effect on the liver function enzymes of normal mice, but decreased them in CYP-treated mice (*p* < 0.001, *p* < 0.0001).

### 3.4. Maca Polysaccharide Reduces Oxidative Stress

MDA is a sign of lipid peroxidation and reflects the severity of oxidative stress and attack of free radicals on cells. SOD, GSH-Px, and CAT are good antioxidant enzymes; they prevent the harmful effects of oxidative stress, and their oxidative activities represent the level of antioxidant capacity of MP. MP increased the levels of these antioxidant enzymes and inhibited increases in MDA levels in CYP-treated mice after administration, with no significant changes in MP Control mice (Figure 5).

### 3.5. Effect of Maca Polysaccharide on the Mitochondrial Energy Metabolism in the Liver Tissues of CYP-Treated Mice

ATPase is located on the biomembrane and maintains the integrity of the membrane and the metabolic functions of the tissue by participating in energy conversion, material metabolism, and information transmission. It can be used as an indicator of tissue damage, metabolic disorder, and recovery ability. The activities of F_0_F_1_-ATPase, Na^+^-K^+^-ATPase, and Ca^2+^-Mg^2+^-ATPase were measured. As shown in Figure 6, there were no significant changes in F_0_F_1_ -ATPase, Na^+^-K^+^-ATPase, and Ca^2+^-Mg^2+^-ATPase level between mice in the Normal Control group and those in the MP Control group. The levels of F_0_F_1_-ATPase and Na^+^-K^+^-ATPase of mice in the MP+CYP group increased significantly (*p* < 0.01, *p* < 0.05), while the level of Ca^2+^-Mg^2+^-ATPase increased but showed no significance compared with mice in the model group. These results suggest that Maca polysaccharide increased the levels of energy metabolism-related enzymes in the liver tissues of CYP-treated mice.

### 3.6. Hepatic Metabolomic Analysis Based on PCA and OPLS-DA

We used UPLC-Q-TOF/MS to collect the data of liver samples, and the results showed that 5535 metabolites were detected in ESI + modes and 6711 metabolites in ESI-modes.

PCA was used to visualize the trend of different processing groups. As shown in Figure 7, QC samples (green squares in the Figures) are closely clustered in the score plot in both ESI+ and ESI− modes. This shows the stability and repeatability of the system and assures that the study is reliable. Then, the differences between the NC group, CYP group, and MP+CYP group were distinguished by the PCA score plot of liver samples. The Normal Control group is shown with red dots, the CYP Model group is shown with blue squares, and the MP+CYP group is shown with orange dots. The PCA data indicate that the structure of hepatic metabolites between the CYP vs. NC group and CYP vs. CYP+MP group were well separated in both ESI+ and ESI− modes. This shows that CYP treatment disturbed the metabolic spectrum of liver tissue, suggesting that endogenous small molecule metabolites had changed.

The existence of QC samples determined in the metabolomic analysis assures the stability and repeatability of the analytical methods and instruments therein. The QC samples show that the response strength and retention time of each peak have good reproducibility, indicating that the instrument had good stability over the entire analysis process. In addition, the experimental data from the metabolomic analysis are stable and reliable.

Supervised OPLS-DA allowed us to identify the potential biomarkers for the observed variation (Figure 8). The results show that the NC group is significantly separated from the CYP Model group (ESI+: R2Y (cum) = (0, 0.97), Q2 (cum) = (0, −0.37); ESI−: R2Y (cum) = (0, 0.97), Q2 (cum) = (0, −0.37)) and the MP+CYP group is significantly separated from the CYP group (ESI+: R2Y (cum) = (0, 0.97), Q2 (cum) = (0, −0.36); ESI−: R2Y (cum) = (0, 0.95), Q2 (cum) = (0, −0.43)). The sample of each group can be completely separated in the PC1 dimension, and the hepatic metabolomics of mice in each group have obvious characteristics.

Furthermore, 200 displacement tests were checked to verify whether the OPLS-DA was overfitted. As shown in Figure 9, all R2 and Q2 values of the mode are lower than the original value, while the Q2 value is close to zero, which indicates that there is no overfitting phenomenon in the OPLS-DA mode. Thus, the results are highly reliable.

### 3.7. Screening and Identification for Potential Biomarkers

In OPLS-DA modes, the basis for selecting potential variables as biomarkers is whether VIP value > 1 and *p* < 0.05. The screening and identification of differential metabolites were compared with online databases METLIN and HMDB. In the metabolomic analysis, 32 metabolites were identified between the NC group and CYP group, which could be considered as endogenous biomarkers induced by CYP. In addition, 31 metabolites showed differences between the CYP group and MP+CYP group. In CYP-treated mice, after administration of MP, the metabolites of 21 recognized biomarkers in the NC group and CYP group recovered significantly, including 12 in ESI+ mode and 6 in ESI-mode. The changing trend of biomarkers between groups is shown in Table 2.

The differential metabolites identified were analyzed by Cluster heat map analysis to make the metabolic differences of mice in different groups more clear. The heat map in Figure 10 directly shows the trends of each biomarker, indicating that the values among the NC group, CYP group, and MP+CYP group increase or decrease relatively.

### 3.8. Metabolic Pathway of Potential Biomarkers

In order to further understand and visualize the possible metabolic pathways of MP administration on CYP-induced hepatotoxicity, we used Metaboanalyst5.0 to analyze metabolic pathways (Figure 11). Supplementation with MP affected glycerophospholipid metabolism, pyrimidine metabolism, the pentose phosphate pathway, lysine degradation, cysteine and methionine metabolism, arachidonic acid metabolism, and pantothenate and CoA biosynthesis.

In addition, we visualized the comprehensive metabolic pathway analysis in relation to differential and potential biomarkers (Figure 12). MP alleviates the abnormal lipid metabolism induced by CYP in mice, mainly including glycerophospholipid metabolism and arachidonic acid metabolism. MP administration led to a significant decrease in PC (22:5 (4Z,7Z,10Z,13Z,16Z)/20:5 (5Z,8Z,11Z,14Z,17Z)), PC (16:1 (9Z)/16:1 (9Z)), and LysoPC (20:4 (8Z,11Z,14Z,17Z)) and an increase in glycerophosphocholine and lysoPA (18:1 (9Z)/0:0). PCs and 8,9-DiHETrE related to arachidonic acid metabolism were also identified. CYP injection caused a decrease in orotidine, uridine, and uracil and an increase in deoxycytidine, and the destruction of the pyrimidine pathway may affect nucleotide biosynthesis, resulting in liver injury. MP administration reversed their concentrations, thus playing a protective role in the liver. CYP leads to a compensatory increase in ribose 1-phosphate, which is the precursor of 5-phosphate-alpha-d-ribose 1-diphosphate (PRPP). PRPP further reacts to produce sedoheptulose 7-phosphate, which promotes the formation of saccharine; saccharopine was also identified. MP decreased them. CYP significantly increased the amino acid metabolism disorders of cysteine and methionine metabolism (L-methionine and S-adenosylhomocysteine) and lysine degradation (saccharopine and 4-trimethylammoniobutanoic acid) in hepatotoxicity mice, while MP significantly reversed these markers.

### 3.9. Correlation Analysis between Potential Biomarkers and Biochemical Indices

The R Programming Language (4.1.0) was used to carry out correlation analysis between potential biomarkers and biological indicators. The indicators included SOD, MDA, GSH-Px, CAT, F_0_F_1_ -ATPase, Na^+^-K^+^-ATPase, and Ca^2+^-Mg^2+^-ATPase. r > 0.5 or r < −0.5 were considered to indicate correlation (Figure 13).

The results show a certain correlation between potential biomarkers. For example, glycerophosphocholine is positively correlated with 4-trimethylammoniobutanoic acid and uracil (r = 0.58, 0.56) and negatively correlated with ribose 1-phosphate and sedoheptulose (r = −0.56, −0.57). 8,9-DiHETrE is positively correlated with ribose 1-phosphate and sedoheptulose (r = 0.61, 0.56), which means glycerophospholipid metabolism is correlated with the pentose phosphate pathway and pyrimidine metabolism.

Moreover, the correlation between potential biomarkers and biochemical indices is also obvious. Glycerophosphocholine, 4-trimethylammoniobutanoic acid, and uracil are positively correlated with SOD, GSH-Px (r = 0.64, 0.77; r = 0.55, 0.57; r = 0.65, 0.66), and LysoPA (18:1 (9Z)/0:0). Uridine and L-methionine are negatively correlated with MDA (r = −0.66; r = −0.73; r = −0.65). This reflects that ameliorated oxidative stress is related to these potential biomarkers regulated by MP in CYP-treated mice. The antioxidative indices are quite positively correlated with energy metabolism indices.

### 3.10. Maca Polysaccharide Activates Protective Antioxidant Mechanisms via Keap1-Nrf2 Signaling after Cyclophosphamide Chanllenge

The mRNA and protein expression levels of Nrf2, Keap1, HO-1, Gclc, and Nqo1 in the liver tissues of mice were measured (Figure 14 and Figure 15). Compared with mice in the control group, Nrf2, HO-1, Gclc, and Nqo1 levels decreased while the level of Keap1 significantly increased in the CYP Model group. On the other hand, the levels of Nrf2, HO-1, Gclc, and Nqo1 enzymes of Maca polysaccharide-supplemented mice increased, and the level of Keap1 decreased.

## 4. Discussion

### 4.1. MP Alleviates Cyclophosphamide-Induced Hepatotoxicity in Mice by Ameliorating Oxidative Stress and Improving Energy Metabolism

The combination of dietary antioxidants and chemotherapy or therapeutic drugs to reduce harmful side effects and oxidative organ damage after administration has become the focus of attention [31,32]. MP can reduce the inflammatory cell infiltration of liver tissue, central venous congestion, and hepatocyte necrosis caused by cyclophosphamide; inhibit weight loss; and improve the activities of liver metabolic enzymes (ALT and AST). MP also improves the activity of antioxidant enzymes and the ability to directly scavenge oxygen free radicals, and may alleviate lipid oxidative damage through antioxidation.

It is known that most liver diseases are closely associated with the depletion of mitochondrial energy production [33]. Na^+^-K^+^-ATPase and Ca^2+^-Mg^2+^-ATPase are the two most important ion pump enzymes in ATPase. They can transport ions in and out of cell membranes at the inverse concentration gradient. F_0_F_1_-ATPase is one of the five complex enzymes (complex enzyme V) located in the inner respiratory chain of mitochondria. It generates ATP from ADP and phosphate by using the chemical gradient of proton pumping into the inner space of mitochondria [34,35]. ATPases are considered as important indicators of cell metabolic disorders [36,37]. MP supplementation could inhibit decreases in the two ion pumps and enhance the expression of F_0_F_1_-ATPase, thereby promoting hepatocytes’ energy metabolism.

### 4.2. MP Regulates the Potential Metabolic Markers and Pathways Associated with CYP-Induced Hepatotoxicity

MP alleviates the abnormal lipid metabolism induced by CYP in mice, mainly including glycerophospholipid metabolism and arachidonic acid metabolism. Although the impact value of arachidonic acid metabolism pathway was small, as a supplementary reaction of glycerophospholipid metabolism, it is still important in the process of lipid metabolism [38], which should be paid sufficient attention. Acetyl CoA is essential in acetylation reactions, so pantothenate and CoA biosynthesis also have important contributions in the metabolic process [39].

Phospholipids are important in signal transduction and metabolism, and they are also the main components of cell membranes [40]. When oxidative stress occurs, free radicals can activate phospholipase A2 (PLA2), and PC hydrolyzes to produce LPC [41]. LPC damages the cell membrane of the liver and further aggravates hepatocyte damage [42]. In this study, MP eliminated peroxide free radicals and inhibited lipid oxidation chain reactions. Our results show that MP protects the structure and function of biofilm from oxidative stress by regulating the glycerophospholipid metabolism pathway.

Studies have shown that pyrimidine metabolism and amino acid metabolism are closely related to liver function, as pyrimidine is synthesized and degraded in the liver [43,44]. Lysine is involved in protein synthesis together with other amino acids in the liver. Homocysteine (HCY) is a sulfur-containing amino acid, which is converted from methionine in protein taken in by the human body. S-adenosylhomocysteine is a derivative of HCY. It is an intermediate product of methionine metabolism into glutathione and S-adenosylmethionine, and S-adenosylmethionine is a natural protective agent against hepatotoxicity [45]. Changes in glucose metabolism during hepatocyte injury are characterized by a relatively enhanced pentose phosphate pathway and glycolysis pathway, with poor oxidative phosphorylation and tricarboxylic acid cycles in severe hepatotoxicity [46]. In this study, MP restored mitochondrial function and thus restored liver energy metabolism by regulating pyrimidine metabolism, amino acid metabolism, and the pentose phosphate pathway.

### 4.3. Maca Polysaccharide Attenuates CYP-Induced Hepatotoxicity via Keap1-Nrf2 Antioxidative Pathway

Nrf2 is a transcription factor that regulates and induces the expression of cellular antioxidant genes and liver detoxifying enzyme genes [47]. Under normal conditions, Nrf2 is sequestered in the cytoplasm in association with its protein inhibitor Keap-1. Under oxidative stress conditions, Keap1 is oxidized or covalently modified to uncouple with Nrf2, enabling Nrf2 to activate and translocate to the nucleus. Nrf2 heterodimerizes with antioxidant response element (ARE) and starts the transcription of phase II detoxifying enzyme; phase III transporter; and antioxidant proteins such as SOD, CAT, HO-1, Gclc, Nqo1 [48,49]. Therefore, the Keap1-Nrf2 signaling pathway is important in the mechanism of resistance to oxidative stress [50,51,52]. In this study, preventive administration of MP could alleviate hepatic oxidative stress caused by CYP by regulating the Keap1-Nrf2 pathway.

As mentioned earlier in the article, there is a complicated relationship between oxidative stress, energy metabolism, metabolic disorders, and hepatotoxicity induced by CYP. Regulating the antioxidant capacity of hepatocytes may affect the internal metabolic level. MP regulates lipid metabolism and amino acid metabolism of hepatocytes through the classic oxidative stress Keap1-Nrf2 signaling pathway. In order to discover potential effects of MP supplementation on oxidative stress and metabolism in the liver, we performed a Pearson correlation analysis (Figure 12).

Glycerophosphocholine promotes the synthesis of cell membranes, and it was positively correlated with SOD and GSH-Px (r = 0.64, 0.77), which indicates the antioxidative effect of MP. The phospholipid PC (22:5 (4Z, 7z, 10z, 13z, 16Z)/20:5 (5Z, 8Z, 11z, 14z, 17Z)) was negatively correlated with GSH-Px (r = −0.56), and the lower level of PC indicated the amelioration of oxidative stress. Sedoheptulose was negatively correlated with GSH-Px (r = −0.64), indicating that MP supplementation regulated the pentose phosphate pathway by ameliorating oxidative stress. Furthermore, the levels of uracil and 4-trimethylammoniobutanoic acid were obviously decreased in the CYP Model group and increased in the MP+CYP group, which is in agreement with the result of oxidative stress. According to the results, MP supplementation could significantly ameliorate oxidative stress and attenuate CYP-induced hepatotoxicity.

## 5. Conclusions

The aim of this study was to investigate the effect of MP as a natural antioxidant against CYP-induced hepatoxicity in mice by ameliorating oxidative stress and improving energy metabolism. Through metabolomic analysis and correlation analysis, we found that the antioxidant and energy metabolism-improving effects of MP are closely related to regulating glycerophospholipid metabolism and the pentose phosphate pathway. In addition, this study represents the first time that possible involvement of the Keap1-Nrf2 pathway has been shown in the antioxidant mechanism of Maca polysaccharide.

## Figures and Tables

**Figure 1 nutrients-14-04264-f001:**
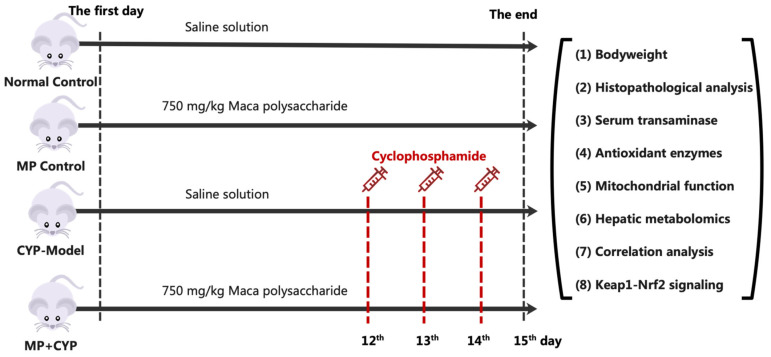
Workflow of the experiment procedure. (1)–(5) were performed for each group; (6)–(8) were performed for NC, CYP Model, and MP+CYP group liver tissues.

**Figure 2 nutrients-14-04264-f002:**
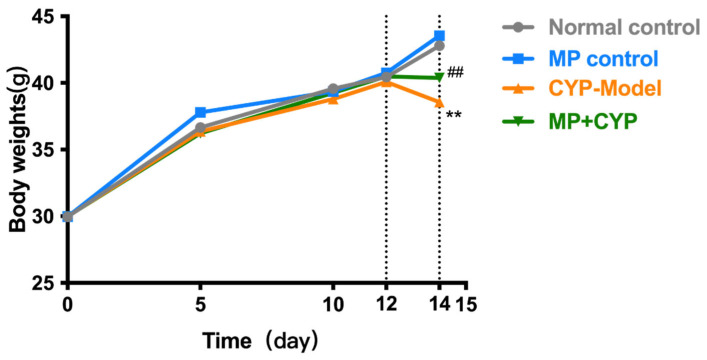
Effects of Maca polysaccharide on body weight of normal and CYP-treated mice (*n* = 12). Compared with Normal Control group: ** *p* < 0.01; Compared with CYP Model group: ^##^ *p* < 0.01. Data are expressed as means ± SEM.

**Figure 3 nutrients-14-04264-f003:**
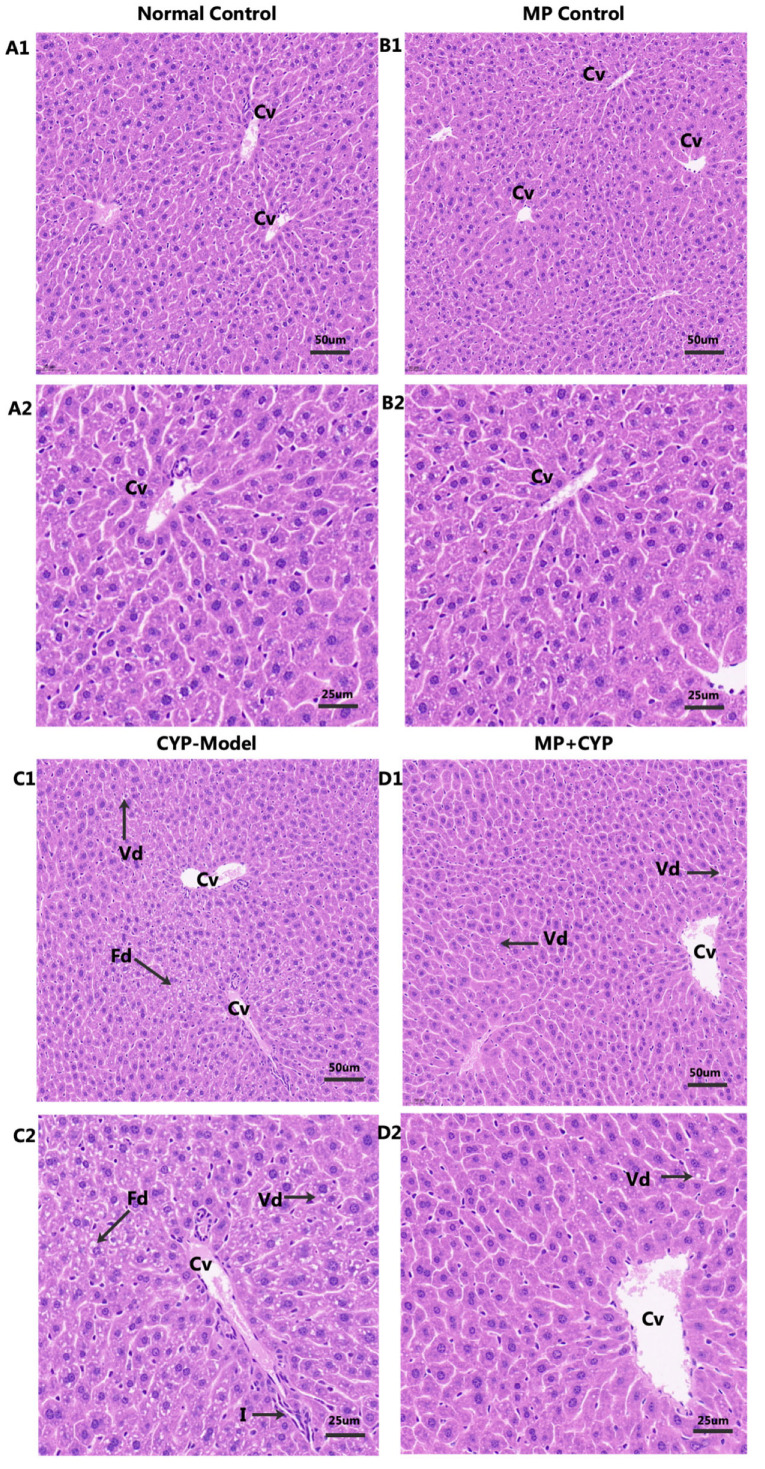
Histopathological observation of liver slices in the Normal Control, MP Control, CYP Model, and MP+CYP groups. Central veins (Cv) were clearly visible in all groups. The Normal Control group (**A1** and **A2**) and MP Control group (**B1** and **B2**) showed normal hepatic chords and lobules. The livers of the CYP Model (**C1** and **C2**) and MP+CYP (**D1** and **D2**) groups showed pathological changes of varying degrees. The CYP Model group (**C1** and **C2**) showed vacuolar degeneration (Vd, arrow), fatty degeneration (Fd, arrow), and inflammatory cellular infiltration (I, arrow). In addition, the MP+CYP group (**D1** and **D2**) showed slight vacuolar degeneration (Vd, arrow). The images depict staining with H&E at 200× and 400× magnification.

**Figure 4 nutrients-14-04264-f004:**
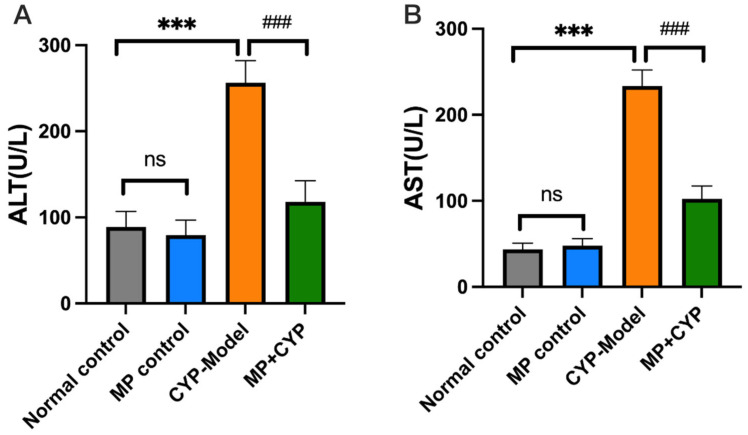
Levels of serum transaminase in different groups of mice (*n* = 12). (**A**) Alanine aminotransferase (ALT); (**B**) Aspartate aminotransferase (AST). *** *p* < 0.0001 compared with Normal Control group; ^###^
*p* <0.0001 compared with CYP Model group. Data are expressed as means ± SEM.

**Figure 5 nutrients-14-04264-f005:**
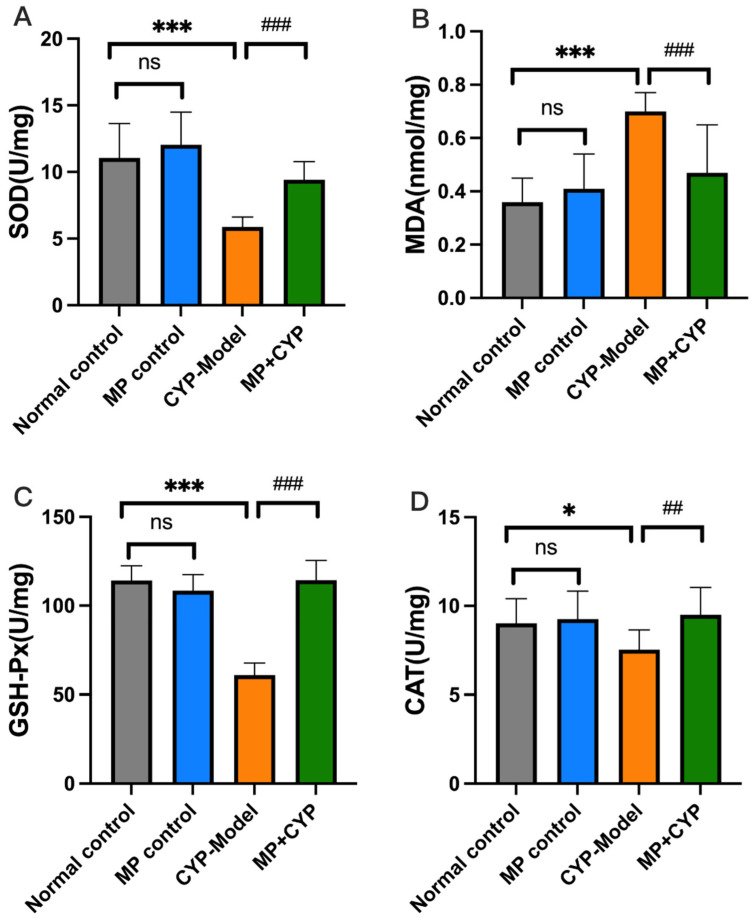
Effect of Maca polysaccharide on antioxidant enzymes in liver tissue of CYP-treated mice (*n* = 12). (**A**) SOD; (**B**) MDA; (**C**) GSH-Px; (**D**) CAT. * *p* < 0.05, *** *p* < 0.0001 compared with Normal Control group; ^##^
*p* < 0.01, ^###^
*p* < 0.0001 compared with CYP Model group. Data are expressed as means ± SEM.

**Figure 6 nutrients-14-04264-f006:**
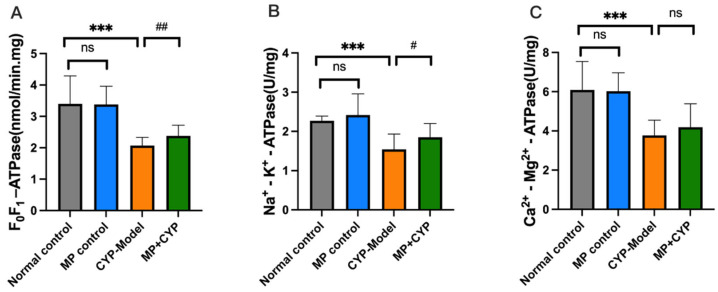
Effect of Maca polysaccharide on enzyme content of energy metabolism in liver tissue of CYP-treated mice (*n* = 12). (**A**) F_0_F_1_ -ATPase; (**B**) Na^+^-K^+^-ATPase; (**C**) Ca^2+^-Mg^2+^-ATPase. *** *p* < 0.0001 compared with Normal Control group; ^#^
*p* < 0.05, ^##^ *p* < 0.01 compared with CYP Model group. Data are expressed as means ± SEM.

**Figure 7 nutrients-14-04264-f007:**
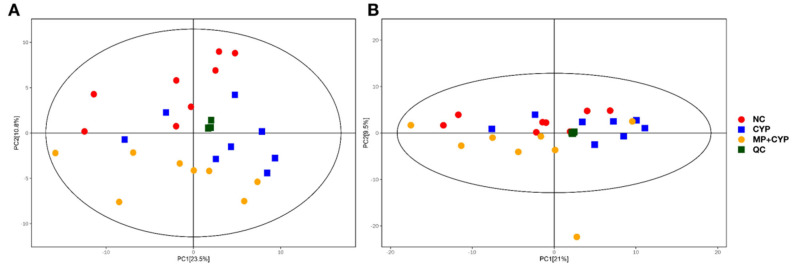
Score scatter plot for PCA model TOTAL with QC. The Normal Control group is shown with red dots, the CYP Model group is shown with blue squares, the MP+CYP group is shown with orange dots, and the QC samples are shown with green squares. (**A**) All samples in ESI+ mode. (**B**) All samples in ESI− mode.

**Figure 8 nutrients-14-04264-f008:**
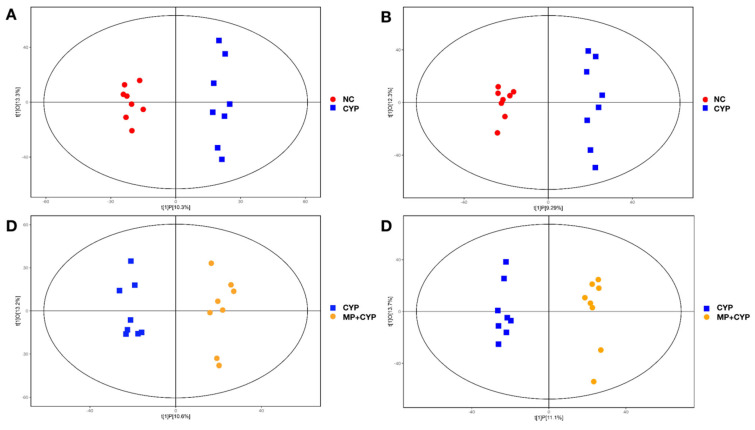
The OPLS-DA score plots. (**A**) NC and CYP Model groups in ESI + mode; (**B**) NC and CYP Model groups in ESI− mode; (**C**) CYP Model and MP+CYP groups in ESI+ mode; (**D**) CYP Model and MP+CYP groups in ESI− mode.

**Figure 9 nutrients-14-04264-f009:**
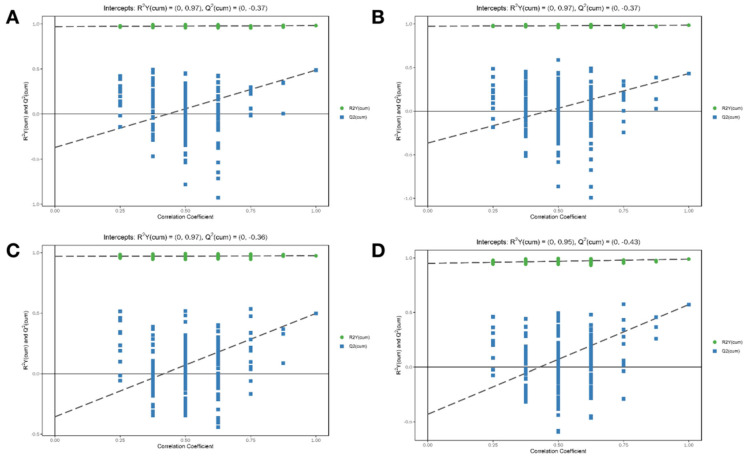
Permutation tests of the OPLS-DA model. (**A**) Normal Control group vs. CYP Model group (ESI+); (**B**) Normal Control group vs. CYP Model group (ESI−); (**C**) CYP Model group vs. MP+CYP group (ESI+); (**D**) CYP Model group vs. MP+CYP group (ESI−).

**Figure 10 nutrients-14-04264-f010:**
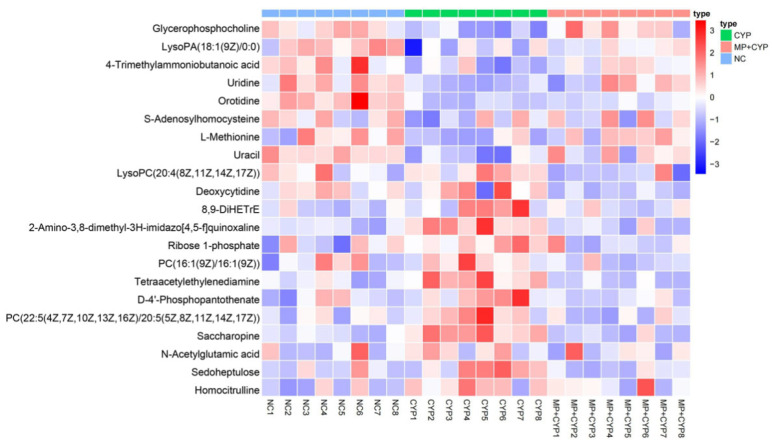
Cluster heat map of differential metabolites.

**Figure 11 nutrients-14-04264-f011:**
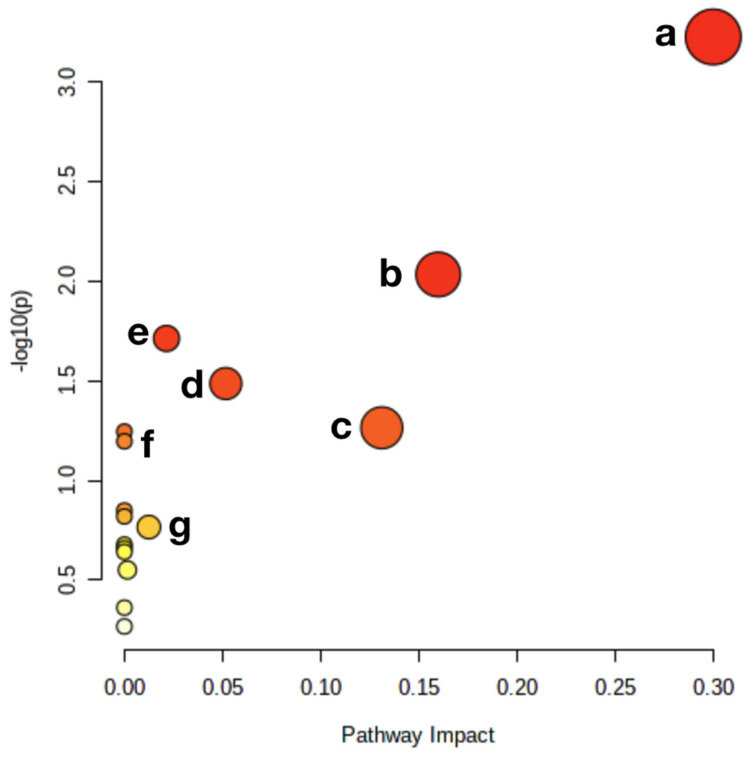
Metabolic pathway analysis. a: Glycerophospholipid metabolism; b: Pyrimidine metabolism; c: Cysteine and methionine metabolism; d: Lysine degradation; e: Pantothenate and CoA biosynthesis; f: Pentose phosphate pathway; g: Arachidonic acid metabolism.

**Figure 12 nutrients-14-04264-f012:**
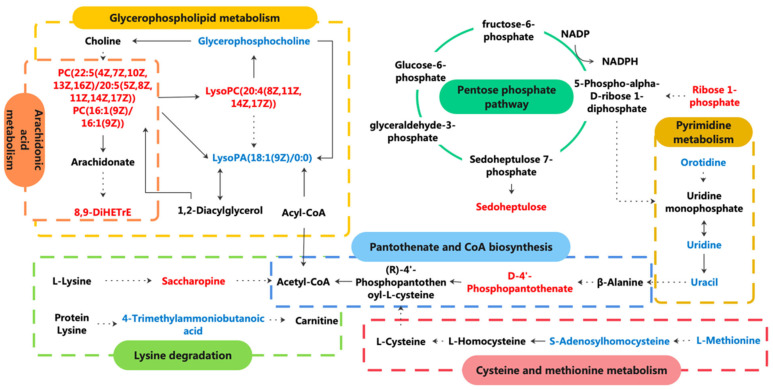
Comprehensive metabolic pathway analysis of potential biomarkers from metabolomics. Metabolites with red markers suggest a significant increase (NC group vs. CYP group) and further decrease in the MP+CYP group. Metabolites with blue markers suggest a decrease in the CYP group and increase in the MP+CYP group.

**Figure 13 nutrients-14-04264-f013:**
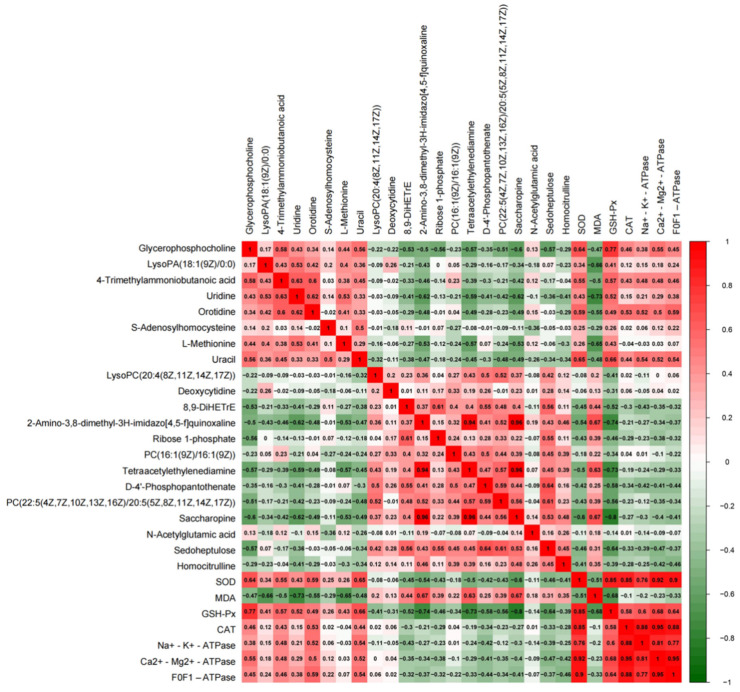
Correlation analysis between potential biomarkers and biochemical indices.

**Figure 14 nutrients-14-04264-f014:**
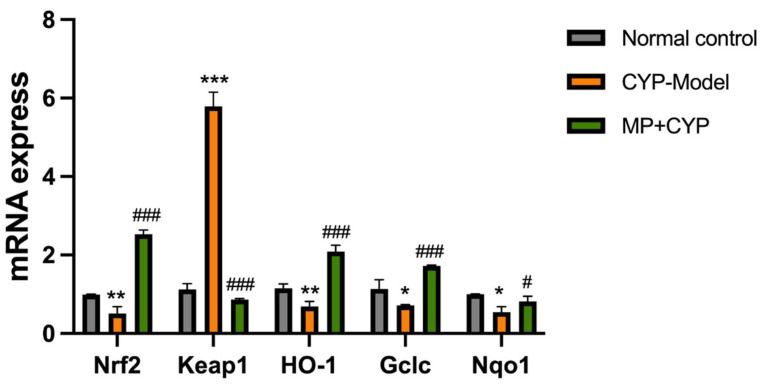
Effect of Maca polysaccharide on mRNA of Keap1-Nrf2 signaling in CYP-treated mice (*n* = 8). * *p* < 0.05, ** *p* < 0.01, *** *p* < 0.0001 compared with Normal Control group; # *p* < 0.05, ### *p* < 0.0001 compared with CYP Model group. Data are expressed as means ± SEM.

**Figure 15 nutrients-14-04264-f015:**
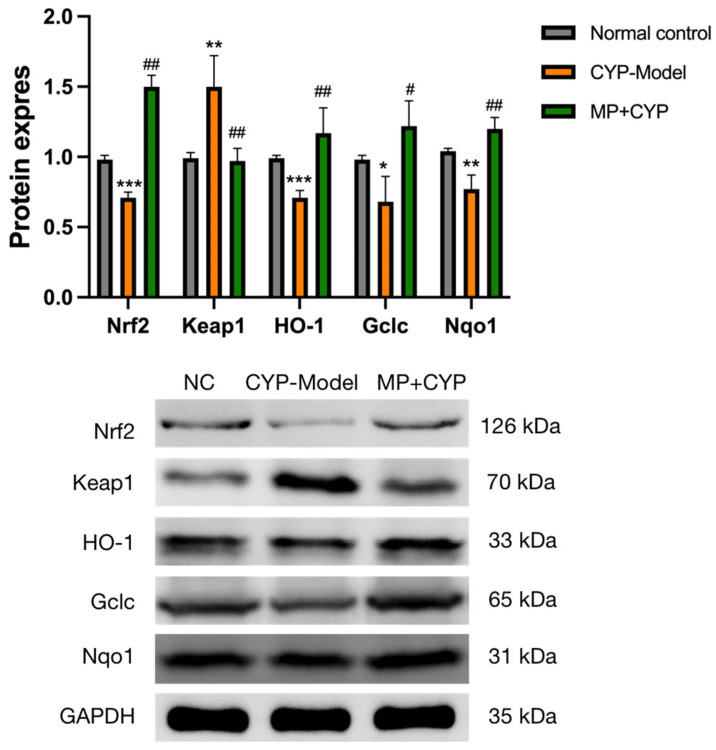
Effect of Maca polysaccharide on protein expression of Keap1-Nrf2 signaling in CYP-treated mice (*n* = 3). Protein expression of Nrf2, Keap1, HO-1, Gclc, and Nqo1 in the liver. * *p* < 0.05, ** *p* < 0.01, *** *p* < 0.0001 compared with Normal Control group; # *p* < 0.05, ## *p* < 0.01 compared with CYP Model group. Data are expressed as means ± SEM.

**Table 1 nutrients-14-04264-t001:** Primer sequences used for quantitative RT-qPCR.

Genes	Forward (5′–3′)	Reverse (5′–3′)
Nrf2	CCTTTGGAGGCAAGACATAGATC	CATCTACAAATGGGAATGTCTCTG
Keap1	GAGATATGAGCCAGAGCGGGA	AACTGGTCCTGCCCATCGTAG
HO-1	AACTTTCAGAAGGGTCAGGTGTC	CTCCTCAGGGAAGTAGAGTGGG
Gclc	CTGTAGATGATAGAACACGGGAGG	GAGATGAGCAACGTGCTGTGC
Nqo1	GCGAGAAGAGCCCTGATTGT	AGATGACTCGGAAGGATACTGAAA
GAPDH	CCTCGTCCCGTAGACAAAATG	TGAGGTCAATGAAGGGGTCGT

**Table 2 nutrients-14-04264-t002:** Differential metabolites in the liver and their change trends among groups.

No.	Differential Metabolites	Mode	CYP vs. NC			MP+CYP vs. CYP		
			VIP	FC	Trend	VIP	FC	Trend
1	Glycerophosphocholine	ESI+	1.66	0.80	↓ *	1.08	1.08	↑ *
2	LysoPA(18:1(9Z)/0:0)	ESI−	1.2	0.71	↓ *	1.69	1.27	↑ *
3	4-Trimethylammoniobutanoic acid	ESI+	1.11	0.72	↓ *	1.60	1.42	↑ *
4	Uridine	ESI+	2.07	0.82	↓ **	1.29	1.18	↑ *
5	Orotidine	ESI−	2.22	0.71	↓ ***	1.37	1.37	↑ *
6	S-Adenosylhomocysteine	ESI+	2.05	0.75	↓ *	1.06	1.30	↑ *
7	L-Methionine	ESI+	1.58	0.88	↓ *	1.70	1.17	↑ *
8	Uracil	ESI−	1.82	1.24	↓ *	1.12	1.02	↑ *
9	LysoPC(20:4(8Z,11Z,14Z,17Z))	ESI+	1.95	1.67	↑ *	1.71	0.64	↓ *
10	Deoxycytidine	ESI−	1.84	0.76	↑ *	1.20	0.76	↓ *
11	8,9-DiHETrE	ESI−	1.59	1.63	↑ *	1.16	0.70	↓ *
12	2-Amino-3,8-dimethyl-3H-imidazo [4,5-f]quinoxaline	ESI+	2.45	2.24	↑ ***	2.37	0.43	↓ ***
13	Ribose 1-phosphate	ESI−	1.47	1.33	↑ *	1.58	0.74	↓ *
14	PC(16:1(9Z)/16:1(9Z))	ESI+	1.19	2.23	↑ *	1.57	0.36	↓ *
15	Tetraacetylethylenediamine	ESI+	2.14	2.25	↑ ***	2.56	0.35	↓ ***
16	D-4′-Phosphopantothenate	ESI+	1.38	1.53	↑ *	1.67	0.62	↓ *
17	PC(22:5(4Z,7Z,10Z,13Z,16Z)/20:5(5Z,8Z,11Z,14Z,17Z))	ESI+	1.45	1.29	↑ *	1.75	0.75	↓ *
18	Saccharopine	ESI−	2.55	2.16	↑ ***	2.47	0.40	↓ ***
19	N-Acetylglutamic acid	ESI−	1.77	1.27	↑ *	2.33	0.68	↓ ***
20	Sedoheptulose	ESI+	1.27	1.79	↑ *	1.99	0.42	↓ **
21	Homocitrulline	ESI+	1.71	1.36	↑ *	1.77	0.66	↓ *

* *p* < 0.05, ** *p* < 0.01, *** *p* < 0.0001. The arrow (↑ and ↓) means the changing trend of biomarkers between groups.

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
