# Peer review of "Antioxidative and Energy Metabolism-Improving Effects of Maca Polysaccharide on Cyclophosphamide-Induced Hepatotoxicity Mice via Metabolomic Analysis and Keap1-Nrf2 Pathway"

_nutrients, 2022, doi:10.3390/nu14204264_

Round 1

Reviewer 1 Report

In their paper, Fei et al. discuss the role of Maca polysaccharide in cyclophosphamide-induced hepatotoxicity in mice. They used a variety of approaches and concluded that Maca polysaccharide has an antioxidant effect and improves metabolism in mice.

I think that the work was done following a scientifically sound criterion, but there is a need for revision.

In general, the introduction, discussion and conclusion require revision. References are not always congruent. Improve the aim of the work. Some bibliographic data included in discussion should be moved to the introduction. Remove the results from the discussion and make the discussion more incisive.

 Below is a list of changes to be made:

Write species names in italics even in references

Check spaces between words and typos

Always write Maca the same way, even in references

Check the abbreviations used

Materials and Methods:

Always indicate more precisely the kits used and the manufacturers and add references for protocols used

Par. 2.4: group I and III; how do the mice receive the saline solution?

Par 2.12: lines 191 and 192 what do the abbreviations 3UG and 50UGL stand for?

Figure 1: separate the measurements made on the samples from the diagram (parentheses or other) and indicate in the caption that these activities are performed for each type of sample. Match the days on which injections are given with the syringe design

Figure 3: in caption replace arrowhead with arrow, add magnification bar, magnification is too small to visualize vacuoles well, and fatty degeneration

Figure 7 and Results: better describe the data of scatter plot for PCA

Figure 12: better describe the data in the Results

Figures 14 an 15: why is the data for Maca control missing?

Author Response

In their paper, Fei et al. discuss the role of Maca polysaccharide in cyclophosphamide-induced hepatotoxicity in mice. They used a variety of approaches and concluded that Maca polysaccharide has an antioxidant effect and improves metabolism in mice.

I think that the work was done following a scientifically sound criterion, but there is a need for revision.

Point 1: In general, the introduction, discussion and conclusion require revision. References are not always congruent. Improve the aim of the work. Some bibliographic data included in discussion should be moved to the introduction. Remove the results from the discussion and make the discussion more incisive.

Response 1: Thank you for your review. The introduction, discussion and conclusion have been revised carefully. Some bibliographic data included in discussion have been moved to the introduction, and the results from the discussion have been removed. In addition, more references were cited relevant to the research. 

 Below is a list of changes to be made:

Point 2: Write species names in italics even in references

Response 2: Thank you for your review. The species names in references have been written in italics.

Point 3: Check spaces between words and typos

Response 3: Thank you for your review. English has been further proofread.

Point 4: Always write Maca the same way, even in references

Response 4: Thank you for your review. I have written Maca the same way in the whole article, even in references.

Point 5: Check the abbreviations used

Response 5: Thank you for your review. The abbreviations used have been checked.

Materials and Methods:

Point 6: Always indicate more precisely the kits used and the manufacturers and add references for protocols used

Response 6: Thank you for your review. The kits used and the manufacturers were supplemented as required, and references of the protocols used in the methods have been cited.

Point 7: Par. 2.4: group I and III; how do the mice receive the saline solution?

Response 7: Thank you for your review. Mice received saline by oral gavage in group I and III. It has been described and supplemented in the Experimental design part.

Point 8: Par 2.12: lines 191 and 192 what do the abbreviations 3UG and 50UGL stand for?

Response 8: Thank you for your review. I’m sorry they are a spelling errors here. It should be: 3 ug of total RNA was retrieved and amplified by PCR in 50 UL reaction system according to the kit instructions.

Point 9: Figure 1: separate the measurements made on the samples from the diagram (parentheses or other) and indicate in the caption that these activities are performed for each type of sample. Match the days on which injections are given with the syringe design

Response 9: Thank you for your review. Figure 1 has been modified to make it clearer.

Point 10: Figure 3: in caption replace arrowhead with arrow, add magnification bar, magnification is too small to visualize vacuoles well, and fatty degeneration

Response 10: Thank you for your review. Magnification bars have been added to the pictures and a 400× magnification have been added, in order to visualize vacuoles well, and fatty degeneration.

Point 11: Figure 7 and Results: better describe the data of scatter plot for PCA

Response 11: Thank you for your review. The data of scatter plot for PCA in Figure 7 and Results were redescribed:

PCA is used to visualize the trend of different processing groups. As shown in Figure 7, QC samples (green squares showed in the Figures) are closely clustered in the score plot in both ESI + and ESI - modes. It showed the stability and repeatability of the system and make sure that the study is reliable. Then, the differences between the NC group, CYP group and MP + CYP group were distinguished by the PCA score plot of liver samples. The Normal Control group were shown in red dots, the CYP-Model group were shown in blue squares, the MP+ CYP group were shown in orange dots. The PCA data indicate that the structure of hepatic metabolites between the CYP vs. NC group and CYP vs. CYP+MP group were well separated in both ESI+ and ESI - mode. It showed that CYP treatment disturbed the metabolic spectrum of liver tissue, and suggested that endogenous small molecule metabolites had changed.

The existence of QC samples in metabolomic analysis can ensure the stability and repeatability of the analytical methods and instruments in the metabolomics analysis. The QC samples showed that the response strength and retention time of each peak had good reproducibility, indicating that the instrument had good stability in the entire analysis process. And the experimental data in the metabolomics analysis was stable and reliable.

Figure 7. Score scatter plot for PCA model TOTAL with QC. The Normal Control group were shown in red dots, the CYP-Model group were shown in blue squares, the MP+ CYP group were shown in orange dots and the QC were shown in green squares. Figure 7A: the PCA score plots of the Normal Control, CYP-Model and MP+CYP groups in positive ion mode (ESI +). Figure 7B: the PCA score plots of the Normal Control, CYP-Model and MP+CYP groups in negative ion mode (ESI -).

Point 12: Figure 12: better describe the data in the Results

Response 12: Thank you for your review. The Results of Figure 12 was redescribed:

In addition, it was visualized the comprehensive metabolic pathway analysis related to differential and potential biomarkers in Figure 12. MP alleviates the abnormal lipid metabolism induced by CYP in mice, mainly including glycerophospholipid metabolism and arachidonic acid metabolism. MP administration led to a significant decrease of PC(22:5(4Z,7Z,10Z,13Z,16Z)/20:5(5Z,8Z,11Z,14Z,17Z)), PC(16:1(9Z)/16:1(9Z)) and LysoPC(20:4(8Z,11Z,14Z,17Z)), while an increase of Glycerophosphocholine and LysoPA(18:1(9Z)/0:0). PCs and 8,9-DiHETrE related to arachidonic acid metabolism were also identified. CYP injection caused the decrease of orotidine, uridine, uracil and the increase of deoxycytidine, and the destruction of pyrimidine pathway may affect nucleotide biosynthesis, resulting in liver injury. MP administration reversed the concentration of them, thus playing a protective role in the liver. CYP leads to the compensatory increase of ribose 1-phosphate, which is the precursor of 5-phosphate-alpha-d-ribose 1-diphosphate (PRPP). PRPP further reacts to produce sedoheptulose 7-phosphate, which promotes the formation of saccharine, and also saccharopine was identified. MP decreased them. CYP significantly increased the amino acid metabolism disorders of cysteine and methionine metabolism (L-Methionine and S-Adenosylhomocysteine) and lysine degradation (Saccharopine and 4-Trimethylammoniobutanoic acid) in hepatotoxicity mice, and MP significantly reversed these markers.

Point 13: Figures 14 an 15: why is the data for Maca control missing?

Response 13: Thank you for your review. The results showed that MP did not change the indexes such as MDA,SOD, GSH-Px and CAT(related to oxidative stress) and F0F1 -ATPase, Na+-K+-ATPase and Ca2+-Mg2+-ATPase (related to energy metabolism) in normal mice, so we didn’t detected the PCR and WB of Keap1-Nrf2 pathway markers.

Reviewer 2 Report

This study shows that Maca (Lepidium meyenii) polysaccharide could reduce a liver tissue damage, such as an oxidative stress, caused by cyclophosphamide, an anti-tumor or an immunosuppressive agent. The reviewer appreciated that the combination treatment with the polysaccharide was effective on alleviating a side effect of the agent. Actually, the substantial data are appeared in this article. However, there would remain a few weaknesses. They are;

1. There is few information about characteristics of a maca polysaccharide. The molecular size and the mode of bind should be described as well as sugar component.

2. Acrolein should have been determined in this study. Because it plays a key role in side effects of cyclophosphamide. In addition, the authors have suggested that an anti-oxidative effect related to ameliorating the side effect. And, acrolein is a peroxidation product.

Author Response

This study shows that Maca (Lepidium meyenii) polysaccharide could reduce a liver tissue damage, such as an oxidative stress, caused by cyclophosphamide, an anti-tumor or an immunosuppressive agent. The reviewer appreciated that the combination treatment with the polysaccharide was effective on alleviating a side effect of the agent. Actually, the substantial data are appeared in this article. However, there would remain a few weaknesses. They are;

Point 1: There is few information about characteristics of a maca polysaccharide. The molecular size and the mode of bind should be described as well as sugar component.

Response 1: Thank you for your review. The characteristics of a maca polysaccharide such as the molecular size and the mode of bind have been described in the article (2.1 Source of Materials). 

Lepidium meyenii (Maca) powder was provided by Yantai Xinshidai Health Industry Co., Ltd. (Shandong, China), the lot number was 20160316. The dried Maca powder was extracted twice by water at 20:1 ml/mg liquid to solid ratio. The extraction conditions used are that the temperature is 70 ℃ for 1.5 h, and the columns used in the purification process are DEAE cellulose chromatography column and Sephadex G-100 gel column. The extraction of MP could reach 9.88 mg/g for further separation, and the purity of total sugar and protein were 75.42% and 7.73% respectively. A polysaccharide MP with a molecular weight of 10.6 kDa was isolated from Maca, and MP was isolated by the method of Zha [14], which is consisted of rhamnose (Rha), arabinose (Ara), glucose (Glc) and galactose (Gal), and the molar ratio was 1.81: 6.85:1:3.21. Arabinose was the main ingredient in MP. MP is mainly composed of →4)-α-D-Glcp-(1→, →6)-α-D-Glcp-(1→, →3)-α-D-Glcp-(1→, and β-D-Araf-(1→, with branching at O-6 of →4,6)-α-D-Glcp-(1 → [14].

Point 2: Acrolein should have been determined in this study. Because it plays a key role in side effects of cyclophosphamide. In addition, the authors have suggested that an anti-oxidative effect related to ameliorating the side effect. And, acrolein is a peroxidation product.

Response 2: Thank you for your review. CYP is metabolized mainly in the liver by cytochrome P450 system into acrolein which is the proximate toxic metabolite. But acrolein is a kind of active α,β unsaturated aldehyde, characterized by high activity, high volatility and thermal instability. Thus, it is insensitive to determine concentration of acrolein in the body. While after intraperitoneal administration of 60mg CYP/kg bodyweight three consecutive days could induce hepatotoxicity. The results of Model group in the article also showed hepatocyte necrosis, congestion in the center of liver lobule, and increase of aminotransferase. We hope that in the next experimental study, we will find a better method to detect the concentration of acrolein and make the experimental data more convincing.

Round 2

Reviewer 2 Report

The reviewer appreciates the response and efforts which has been done by the authors to enhance the quality of the work. However, the description of the marks about P value should be added in every figure, Fig. 4 onward, as well.

Then, after the check and correction of misspelling is done, the manuscript would be acceptable.

Author Response

Thank you for your review. The description of the marks about P value have been added in every figure.